# HR-MAS NMR Metabolomics Profile of Vero Cells under the Influence of Virus Infection and nsP2 Inhibitor: A Chikungunya Case Study

**DOI:** 10.3390/ijms25031414

**Published:** 2024-01-24

**Authors:** Rafaela dos S. Peinado, Lucas G. Martins, Carolina C. Pacca, Marielena V. Saivish, Kelly C. Borsatto, Maurício L. Nogueira, Ljubica Tasic, Raghuvir K. Arni, Raphael J. Eberle, Mônika A. Coronado

**Affiliations:** 1Multiuser Center for Biomolecular Innovation, Department of Physics, Institute of Biosciences, Languages and Exact Sciences (Ibilce—UNESP), Sao Jose do Rio Preto, Sao Paulo 15054000, Brazil; rafaela_peinado@outlook.com (R.d.S.P.); kellyborsatto@gmail.com (K.C.B.); raghuvir.arni@unesp.br (R.K.A.); 2Department of Organic Chemistry, Institute of Chemistry, University of Campinas (UNICAMP), Campinas 13083862, Brazil; lgmartins1984@gmail.com (L.G.M.); ljubica@unicamp.br (L.T.); 3Virology Research Laboratory, Medical School of Sao Jose do Rio Preto (FAMERP), Sao Paulo 15090000, Brazil; carolpacca@gmail.com (C.C.P.); marielenasaivish@gmail.com (M.V.S.); mnogueira@famerp.br (M.L.N.); 4Institute of Biological Information Processing IBI-7: Structural Biochemistry, Forschungszentrum Jülich, 52428 Jülich, Germany; 5Institut für Physikalische Biologie, Heinrich-Heine-Universität Düsseldorf, 40225 Düsseldorf, Germany

**Keywords:** arbovirus, Chikungunya, metabolomics, ^1^H-NMR, Vero cells, wedelolactone

## Abstract

The arbovirus Chikungunya (CHIKV) is transmitted by *Aedes* mosquitoes in urban environments, and in humans, it triggers debilitating symptoms involving long-term complications, including arthritis and Guillain-Barré syndrome. The development of antiviral therapies is relevant, as no efficacious vaccine or drug has yet been approved for clinical application. As a detailed map of molecules underlying the viral infection can be obtained from the metabolome, we validated the metabolic signatures of Vero E6 cells prior to infection (CC), following CHIKV infection (CV) and also upon the inclusion of the nsP2 protease inhibitor wedelolactone (CWV), a coumestan which inhibits viral replication processes. The metabolome groups evidenced significant changes in the levels of lactate, myo-inositol, phosphocholine, glucose, betaine and a few specific amino acids. This study forms a preliminary basis for identifying metabolites through HR-MAS NMR (High Resolution Magic Angle Spinning Nuclear Magnetic Ressonance Spectroscopy) and proposing the affected metabolic pathways of cells following viral infection and upon incorporation of putative antiviral molecules.

## 1. Introduction

Chikungunya virus (CHIKV) is a reemerging arbovirus (arthropod-borne virus) from the *Togaviridae* family and *Alphavirus* genus mainly transmitted by mosquitoes of the *Aedes* genus (*Ae. aegypti* and *Ae. albopictus*), which along with other arboviruses such as Dengue, Zika and yellow fever virus are a public global health concern, as they are spread worldwide [1] and are responsible for significant global human morbidity, mortality and recurrent epidemics [2]. The aforementioned viruses belong to neglected tropical diseases (NTD), which, although manageable, affect more and more of the world’s population [3]. Low investment in the already fragile healthcare systems and a lack of potent vaccines or antiviral drugs make it difficult to control the spread of these infectious diseases [4,5].

Acute symptoms of CHIKV infection include high fever, myalgia, arthralgia, skin rashes, nausea, headaches, intense asthenia and conjunctivitis. Eventually, CHIKV may also cause long-term debilitating conditions and complications, such as arthritis, meningoencephalitis, encephalopathy, hepatitis, myocarditis and Guillain-Barré syndrome [6,7].

CHIKV is a positive-sense RNA virus (12 kb) with two open reading frames that encode four nonstructural proteins (nsP1-nsP4), which are primarily responsible for viral replication, transcription and host evasion, and five structural proteins (C—capsid, E3, E2, 6K/TF and E1—envelope), which are encoded by subgenomic RNA in infected cells [8]. Among the nonstructural proteins, nsP2 plays diverse roles in replication, propagation and host gene expression shutdown, displaying proteolytic, NTPase and RNA helicase activities within its C- and N-terminal domains, making nsP2 a potential target for antiviral research [9,10].

Clinically approved vaccines or antiviral drugs are not yet available against CHIKV. Thus, disease prevention still relies solely on vector control and the treatment of symptoms with anti-inflammatory, analgesic and antipyretic drugs. Globally, scientific efforts have focused on the discovery of novel antiviral drugs and the repurposing of existing drugs such as chloroquine [11], ribavirin [12], favipiravir [13], immunotherapy [14] and other inhibitor molecules [15], but further research is vitally important [16]. In this context, wedelolactone (WDL), a coumestan derived from *Eclipta prostata* (*Asteraceae*), has demonstrated potential as an antioxidant [17], anti-inflammatory [18,19,20,21], hepatoprotective [22,23,24] and anticancer [25,26,27,28,29,30] molecule and also as an inhibitor of HIV-I (human immunodeficiency virus type I) [31], hepatitis C virus RNA polymerase [32] and replication of HCMV (human cytomegalovirus) [33]. However, no studies have been reported on its activity toward CHIKV nsP2 nor CHIKV infection in cellular models.

Understanding how viruses hijack the host cellular machinery and how potential antiviral drugs affect cell metabolism when placed in the cellular environment is imperative. HR-MAS ^1^H-NMR metabolomics permits the identification of metabolites with experimental reproducibility and relatively rapid sample preparation, which makes it well-suited for diverse solid/semi-solid sample types [34]. The spectral data obtained represent unique chemical fingerprints that, combined with statistical methods, provide valuable information on metabolic alterations caused by different conditions, such as diseases, pathogens and inhibitor molecules [35,36,37].

In the present study, we explored the inhibitory potential of wedelolactone against the nonstructural protein 2 of CHIKV as well as the metabolic disturbances in model mammalian cultured cells (Vero E6—African green monkey kidney cells). Vero E6 cells are highly permissive and commonly used in viral propagation and viral assays, and while they lack interferon response genes, our primary objective was to explore the impact of the virus’s presence and potential inhibitors on their metabolism. We explored the changes caused by both CHIKV and WDL using HR-MAS ^1^H-NMR spectroscopy methods. We suggest that metabolomics studies of viruses and inhibitors may represent a rapid and efficient approach for unveiling viral and inhibitor mechanisms that serve as the basis for further investigation of metabolic pathways and drug development through omics integration.

## 2. Results

### 2.1. In Vitro Inhibition Assay of the CHIKV ns2 Protease (nsP2) by Wedelolactone

Experiments employing a wide range of WDL concentrations (ranging from 0 to 50 µM) were performed, and the titration of the molecule demonstrated 100% protease inhibition at 25 µM WDL. Using the titration data, the half-maximal inhibitory concentration (IC_50_) of WDL toward nsP2 was determined to be 2.3 ± 0.4 µM (Figure 1). This value is in the same range as previously reported flavonoids and small molecules targeting CHIKV nsP2: hesperetin (2.5 µM) and hesperidin (7.1 µM) [38] and 1,3-thiazolbenzamide derivatives (between 8.3 and 13.1 µM) [39].

### 2.2. Assessment of Wedelolactone Cytotoxicity In Vero E6 Cells

A cytotoxicity assay of WDL was performed on Vero E6 cells to obtain the CC_50_ value. Cells were treated with different concentrations of WDL and DMSO (negative control) for an incubation period of 48 h. Living cells were quantified through the MTT assay by assessing the production and solubilisation of formazan crystals [40]. The results yielded a CC_50_ value of 373.5 ± 154.2 µM, calculated using GraphPad Prism (8.0) from a normalised dose–response curve fitting (Figure 2). Cell viability was also calculated for each concentration used and is available at Appendix A.

### 2.3. Effects of Wedelolactone on CHIKV-Infected Cells

To evaluate the in vitro inhibitory potential of WDL on CHIKV infection, we treated infected cells with different concentrations of WDL (5, 10, 15, 20 and 30 µM) based on our nsP2 inhibition assay results. This was performed to evaluate the antiviral effects of the molecule on postentry steps, including replication, virion assembly and release, as these were the focus of our metabolomics analyses conducted during the postinfection period. After a 24 h period postinfection and treatment, supernatants were collected, and viral titration was performed. We observed a slight decrease (ANOVA, Dunnett’s test, *p* < 0.0001) in viral titres with concentrations of 20 µM and 30 µM, with 0.44 and 0.45 Log_10_ units decreases in viral titres, respectively, compared to the viral progeny titres in controls (6.92 Log_10_ units, corresponding to approximately 10^7^ PFU/mL); see Figure 3. The IC_50_ for WDL in this assay was calculated and determined as 22.9 ± 6.7 µM.

### 2.4. HR-MAS NMR Spectroscopy Applied to the Analysis of Vero E6 Cells

Twenty-nine metabolites were assigned in the ^1^H-NMR spectra of the Vero E6 cell control group (Figure 4, Appendix A). The 2D-HSQC (Appendix A) and 2D-TOCSY experiments were used to identify and assign the metabolites accurately.

Representative NMR spectra of each group with identified metabolites are shown in Appendix A. The T_2_-edited (CPMG) ^1^H NMR spectra highlighted the most relevant differences in the aliphatic region (0.5–4.55 ppm).

### 2.5. Chemometric Analysis

We first used an unsupervised discriminant model (PCA) to explore intrinsic groupings and reduce dimensionality of datasets from the different conditions. The clustering tendency between the control samples and the other groups reflects the overall changes affected by infection or the inhibitor (Appendix A). The postscattering between the first two components of NMR data of CC × CW, PC1 (27.2%) and PC2 (18.7%), presented a net variance of 45.9% (Appendix A). The other groups presented a net variance of 56.1% (CC × CV), 60.9% (CC × CWV) and 60.3% (CV × CWV) (Appendix A), which indicates significant variability before and after the infection and following the addition of the WDL. PCA analysis was performed on the data matrices of all groups, indicating that the control cells and cells treated with the inhibitor (CC and CW) were similar among themselves, and as expected, similarities among the infected cells (CV and CWV) were also observed (Appendix A).

To focus our attention on the influence of the virus and the inhibitor on the metabolomic profiles of the Vero E6 cells, we assessed the significance of the results using a supervised PLS-DA model. As presented in Figure 5a–d, the PLS-DA class discrimination model maximised the difference between the groups and obtained better separation effects than PCA, demonstrating substantial differences in cellular metabolism among the analysed conditions and presenting adequate cross-validation results of the model through values of accuracy, R^2^ and Q^2^ (Table 1).

### 2.6. Metabolite Discrimination

The most significant variables contributing to class discrimination in the PLS-DA models were identified through VIP graphs. Metabolites with VIP scores > 1.5 in the PLS-DA model (Appendix A) and *p* < 0.05 in the *t*-test were selected as differential metabolites. Box and whisker plots show relative concentrations for the most significantly altered metabolites (*p* < 0.05) in cell control (CC, green) and CV (red), CW (blue) and CWV (purple) (Figure 6). False discovery rate (FDR) estimates and the *p*-value threshold are presented in Appendix A.

Six of the twenty-nine identified metabolites showed significant differences between CC × CV; they are proline (δ 4.14), phosphocholine (δ 4.22), myo-inositol (δ 3.52), lactate (δ 1.33), valine (δ 2.28) and phenylalanine (δ 3.11) (Figure 6a). The cells treated with WDL (Figure 6b) presented differences in four metabolite levels: glucose (δ 3.54), betaine (δ 3.89), isoleucine (δ 3.67) and aspartate (δ 2.68). The corresponding CWV samples showed similar tendencies with five significant metabolite differences between the compared groups (CC × CWV): methionine (δ 3.85) lactate (δ 1.33), proline (δ 4.14), betaine (δ 3.89) and aspartate (δ 2.68) (Figure 6c). Lastly, we compared the infected cells (CV) with the infected WDL-treated cells (CWV), and the most significant metabolites were methionine (δ 3.85), glucose (δ 3.54), aspartate (δ 2.68), myo-inositol (δ 3.52), creatine (δ 3.93), proline (δ 2.36) and lactate (δ 1.33) (Figure 6d).

### 2.7. Heat Map Analysis of Metabolites

Hierarchical clustering analyses (HCA) of the PLS-DA data were performed to explore the substantial changes in each metabolite among the groups. The heat maps visually present the relative increase or decrease in differential metabolite levels in Vero E6 cells in each condition described in the previous analyses (Figure 7)**.**

The heat map displayed significant differences and clustering of groups along with differences in metabolites’ relative concentrations, demonstrating the overall metabolic variation in the different conditions. In CHIKV-infected cells, levels of myo-inositol and amino acids (proline, phenylalanine and valine) increased, while decreased levels of phosphocholine and lactate were observed (Figure 7a). Cell treatment with WDL decreased relative levels of aspartate, whereas levels of glucose, betaine and isoleucine increased (Figure 7b). Furthermore, the CHIKV-infected cells treated with WDL showed increased methionine levels, while aspartate, betaine, lactate and proline showed lower intensities (Figure 7c). When comparing the CV × CWV results, we observed a decrease in metabolites that were increased during viral infection (CV); they were myo-inositol and proline. Moreover, similar to the CW samples, an increase in glucose, methionine and lactate and decreased aspartate levels were observed. Interestingly, here we also identified an alteration in creatine, which presented higher levels in CHIKV-infected cells and lower levels when the infected cells were treated with WDL (Figure 7d). Appendix A shows the heat map with all conditions, and the metabolic alterations will be discussed further.

## 3. Discussion

Our results showed that WDL possesses an inhibitory potential towards CHIKV nsP2 protease in vitro in the low µM concentration range (IC_50_ value of 2.3 ± 0.4 µM). Our value is similar to IC_50_ values reported previously for flavonoids targeting CHIKV nsP2 (HST: 2.5 ± 0.4 µM, HSD: 7.1 ± 1.1 µM) [38]. In contrast, we observed a slight antiviral effect in CHIKV-infected Vero cell cultures. A major problem of the development and discovery of potent antiviral molecules is an efficient cellular delivery; molecules with a poor permeability possess a reduced bioavailability [41]. Additionally, the molecules can be stored in cellular compartments where they are not available for the virus target proteins. However, the treatment of CHIKV-infected Vero cells with WDL demonstrated metabolic alterations that we could observe through our metabolomic experiments.

As virus formation depends on the host cell’s metabolic capacity to provide the required low-molecular-weight metabolites (nucleotides, amino acids and fatty acids (FAs)/lipids) and energy in favour of viral infection, the current study speculates about the metabolic changes of Vero E6 cells induced by CHIKV infection and the inclusion of an nsP2 inhibitor (WDL) exclusively through the HR-MAS ^1^H NMR method.

In our findings, we observed a significant reduction in lactate production in infected cells (CV) when compared to control cells (Figure 6a). This stands in contrast to the findings in MAYV-infected cells, where lactate production exhibited an increase 8–10 h postinfection [42], suggesting a dynamic shift in glycolytic activity throughout the infection process. The sustained decrease in lactate production over a prolonged period (24 h postinfection) leads us to hypothesise that glucose may be metabolised into pyruvate, subsequently entering the TCA cycle. Alternatively, it is plausible that lactate is repurposed as a gluconeogenic agent, serving as a vital energy source [43]. It is important to emphasise the role of the TCA cycle in synthesising amino acids, lipids, nucleotides (purine and pyrimidines) and nucleic acids [44,45,46,47,48] and its significant contribution to ATP synthesis, which is crucial to meet the high-energy requirements during infection. Additionally, overall evidence of glycolysis requirement for efficient replication of alphaviruses has been reported for Semliki, Sindbis and Mayaro viruses, where glycolysis inhibitors substantially diminish viral yields, indicating that glycolysis plays a pivotal role during infection [42,49].

In cultured cells, amino acids from culture media are assimilated and synthesised to form the necessary pools crucial for sustaining normal metabolism, biosynthetic activities and overall cellular homeostasis, and they also play a pivotal role during viral infection [50,51,52]. Accordingly, our results showed that valine, phenylalanine and proline were increased during CHIKV infection. These findings demonstrate broad alterations in glycolysis and TCA cycle metabolism to actively support and facilitate the infection process [53] (Figure 6a and Figure 8).

We also observed elevated levels of myo-inositol in response to CHIKV infection (Figure 6a and Figure 8). Myo-inositol serves as a major component of phospholipids and inositol phosphate derivatives, such as phosphatidylinositol [54]. Beyond its primary role in sensing metabolic shifts and energy demands through inositol phosphate signalling pathways [55], there is a presumed involvement in the assembly and maintenance of the integrity of replication complexes along with recruitment of host factors or viral components, as described by Zhang and collaborators [56].

Viral infection also regulates cellular lipid metabolism [57]. Accordingly, our results demonstrated perturbation of phospholipid metabolism with decreased phosphocholine levels during infection (Figure 6a and Figure 8). Choline and phosphocholine are substrates for the synthesis of phosphatidylcholine (PC), which constitutes approximately half of the phospholipid species in most mammalian cells [58] and is a biosynthetic precursor for lipid-signalling molecules and lipoproteins [59]. PC synthesis is enhanced during the infection of numerous viruses, including Dengue virus [60], poliovirus [61] and Brome mosaic virus (BMV), with a high accumulation of PC and choline metabolism enzymes for PC synthesis within the viral replication sites to sustain viral replication complexes (VRCs) and stimulate viral replication [62].

The relevance of understanding the metabolic changes induced by a CHIKV nsP2 protease inhibitor may favour future studies on drug development. The inhibitor is introduced into an intricate environment and undergoes metabolisation by cells. In this condition (CW), alterations in central energy metabolism were observed upon WDL addition. The reduction in aspartate, a critical building block for several biomolecules and a glucogenic amino acid within the TCA cycle [63], may reflect the promotion of gluconeogenesis upon the addition of WDL, a phenomenon further supported by the elevated glucose levels (Figure 6**b** and Figure 8).

Metabolic changes upon adding WDL (CW) also indicates higher levels of betaine (Figure 6**b** and Figure 8). Betaine, derived from choline oxidation, is an essential osmoregulatory metabolite and a cofactor of methylation during the methionine–homo-cysteine cycle [64,65]. Interestingly, betaine acts as an antiviral and anti-inflammatory compound that may exert functions through epigenetic regulation and mitigation of oxidative and ER (endoplasmic reticulum) stress and cell apoptosis [66]. A notable decrease in betaine levels followed by elevated levels of the sulphur-containing amino acid methionine were observed in our CC × CWV analysis (Figure 6c). As previously mentioned, betaine serves as a methyl donor for methionine synthesis through the activity of the betaine homocysteine S-methyltransferase enzyme [67]. The subsequent steps in the methionine cycle generate a pivotal methylating agent, SAM (S-adenosylmethionine), a cofactor essential for various cellular processes [68] (Figure 8). Both methionine and SAM can be helpful in oxidative stress conditions, as induced by CHIKV infection [69], and the betaine–methionine interaction exerts an antioxidant function by maintaining cellular SAM:SAH ratios through increased methionine and SAM synthesis [66,70]. Furthermore, methionine and SAM synthesis are inducers of anabolic pathways such as amino acid and nucleotide biosynthesis to support cells under nutritional stress conditions (i.e., viral infection), which might also be correlated to the presence of WDL in the cellular environment during infection [71].

WDL consistently appears in the literature as possessing antioxidant effects, demonstrated by its ability to diminish levels of reactive oxygen species (ROS) and mitigate oxidative stress damage [21], promoting free radical scavenging [17] and alleviating oxidative stress and mitochondrial dysfunction [72]. Due to the limitations of our study, further research is necessary to elucidate these effects and mechanisms and investigate whether WDL might play a role against oxidative stress and cell damage caused by CHIKV infection.

Lastly, overall metabolic alterations in the CV × CWV analysis represented similar changes induced by the molecule in the CC × CW conditions, possibly enhancing energy production even amidst viral infection. Nevertheless, we noted decreased levels of myo-inositol and proline, aligning with diminished viral production. This observation contrasts with the observed increase in these metabolites to support viral infection, as previously discussed in the CC × CV results. Interestingly, exclusively in this analysis, creatine was found to be significantly decreased in CWV samples (Figure 6d). Creatine synthesis requires glycine, arginine and methionine [73], and creatine and phosphocreatine are known to regulate mitochondrial ATP synthesis during high-energy-demanding periods to help maintain ATP efficiency in cells [74,75,76]. Furthermore, it was elucidated that creatine kinase B (CKB), a pivotal ATP-generating enzyme responsible for modulating ATP levels within subcellular compartments, plays a crucial role in facilitating the efficient replication of the HCV genome and the subsequent propagation of the infectious virus [75]. Specifically, the recruitment of CKB to the HCV replication complex (RC) compartment, characterised by high and fluctuating energy demands, is paramount for the optimised replication of the viral genome. This recruitment is facilitated through its interaction with NS4A, underscoring the significance of this molecular interplay in the intricate process of viral replication [75]. Based on these findings, Parte superior do formuláriowe speculate that the decreased levels observed in CWV when compared to the CV samples may be linked to diminished energetic demands, a consequence of the lower yields in viral progeny.

## 4. Materials and Methods

### 4.1. Cloning, Expression and Purification of CHIKV nsP2^pro^

The codon-optimised cDNA encoding CHIKV nsP2^pro^ (GenBank Protein Accession number AAN05101.1, strain S27-African prototype) was cloned, expressed and purified as described previously [38,77].

### 4.2. Viral Protease Inhibition Assay

The CHIKV nsP2 protease activity assay was performed as described previously [38,77], using DABCYL-Arg-Ala-Gly-Gly-Tyr-Ile-Phe-Ser-EDANS (BACHEM, Bubendorf, Switzerland) as substrate. The inhibition assay was performed in Corning 96-well plates (Merck, Darmstadt, Germany). An amount of 10 µM CHIKV nsP2^pro^ was incubated with 0, 0.05, 0.1, 0.25, 0.5, 1.0, 2.5, 5.0, 10, 25 and 50 µM WDL for 10 min at RT—the measurement started by adding the substrate with a final concentration of 5 µM. The fluorescence intensities were measured at 60 s intervals over 20 min at 37 °C using an Infinite 200 PRO plate reader (Tecan, Männedorf, Switzerland). The excitation and emission wavelengths were 340 nm and 490 nm, respectively.

Activity and inhibition of the investigated protease was calculated using Equation (1):% protease activity: (Intensity of enzyme activity − intensity left after inhibition)/Intensity of enzyme activity(1)

The experiments were performed in triplicate and the results are shown as mean ± standard deviation (SD). Each experiment was performed with freshly purified protein. The half-maximal inhibitor concentration (IC_50_) value of WDL was calculated using GraphPad Prism 5 software (San Diego, CA, USA).

### 4.3. Cell, Virus and Reagents

Vero E6 cells (African green monkey kidney-derived cells—*Cercopithecus aethiops*, clone E6, ATCC C1008) were grown and maintained at 37 °C and 5% CO_2_ in a humidified incubator in Eagle’s Minimum Essential Medium (MEM—Gibco, Waltham, MA, USA) supplemented with 10% (*v*/*v*) foetal bovine serum (FBS—Gibco, Waltham, MA, USA) and the addition of 100 U/mL of penicillin (Hyclone Laboratories, Logan, UT, USA), 0.1 mg/mL of streptomycin and 0.5 µg/mL of amphotericin B (Gibco, Waltham, MA, USA). CHIKV (BHI3762/H804917 access number [78]) stocks were propagated and titrated in Vero cells using plaque-forming units assay (PFU). Wedelolactone (“WDL”, 7-methoxy 5,11,12-trihydroxy-coumestan—Sigma Aldrich, MA, USA, purity ≥ 98%) was dissolved in DMSO (dimethyl sulfoxide, Merck, NJ, USA, 10 mg/mL) and used for the cell viability and antiviral assays as well as in HR-MAS ^1^H NMR sample preparation.

### 4.4. Cytotoxicity Assay

Cell viability was measured by MTT [3-(4,5-dimethylthiazol-2-yl)-2,5-diphenyl tetrazolium bromide] (Sigma Aldrich, MA, USA) assay according to Mosmann [79] with slight alterations. Vero E6 cells at a 10^4^ density were grown in 96-well plates and treated with different concentrations of WDL ranging from 50 to 400 µM for 48 h at 37 °C with 5% CO_2_, and DMSO was used as the vehicle control. After this period, molecule-containing media were removed, and MTT solution at 1 mg/mL was added to each well, incubated for 1 h and later replaced with 100 µL of DMSO for formazan crystals solubilisation. Absorbance was measured at 550 nm by a Spectramax Plus Microplate Reader (Molecular Devices, Sunnyvale, CA, USA). The CC_50_ (cytotoxic effects on 50% of cultured cells) was defined as the concentration that reduced the viability of cells to 50% when compared to controls. The MTT assay was performed in triplicate, and the results were calculated in GraphPad Prism 8 software (San Diego, CA, USA) using a normalised dose–response curve fitting.

### 4.5. Virus Infection and Titration

A CHIKV stock was used to infect Vero E6 cells in a T-75 culture flask (ThermoFisher, Waltham, MA, USA) at 80% confluency. To determine the viral titre, we seeded 10^5^ Vero E6 cells in each well of a 24-well plate 24 h prior to infection. Cells were then infected with 10-fold serial dilutions of CHIKV and incubated at 37 °C for 1 h with gentle agitation every 15 min. The virus inoculum was removed, and overlay media containing 4% *v*/*v* carboxymethylcellulose (CMC—Synth, Sao Paulo, Brazil) and 1% FBS were added to the wells, followed by incubation for two days at 37 °C and 5% CO_2_ in a humidified incubator. Finally, cells were fixated with 10% formaldehyde and stained with 4% violet crystal, and viral plaques were counted to determine the virus titre, expressed as plaque-forming units, PFU/mL.

### 4.6. Antiviral In Vitro Activity Assay

A postinfection treatment assay was performed to evaluate the in vitro antiviral activity of WDL. A 48-well tissue culture plate with 6 × 10^4^ Vero E6 cells/well was prepared and incubated for 24 h at 37 °C in a humidified atmosphere with 5% CO_2_. After that, media were removed, and cells were infected with CHIKV at a MOI (multiplicity of infection) of 0.1 and incubated for 1 h at 37 °C in a humidified atmosphere with 5% CO_2_. The inoculum was removed, five different diluted concentrations of WDL in MEM 1% FBS were added (30, 20, 15, 10 and 5 µM) to the wells and cells were incubated for a 24 h period at 37 °C in a humidified atmosphere with 5% CO_2._ Supernatants were collected, and viral titration was performed and revealed through the plaque formation assay to evaluate plaque reduction and antiviral activity. The assay was performed in triplicate, and data were analysed by a four-parameter dose–response curve fitting using GraphPad Prism 8 software (San Diego, CA, USA).

### 4.7. HR-MAS ^1^H-NMR Sample Preparation

An amount of 6 × 10^6^ Vero E6 cells were seeded in 100 mm culture dishes 24 h prior to infection and treatment. Six independent samples of each of the following treatment conditions were produced: healthy cell control (CC), cell-WDL (CW), cell-CHIKV (CV) and cell-WDL-CHIKV (CWV).

First, cells were infected with CHIKV at 0.1 MOI (as we identified high cell death rates with higher MOIs) and incubated for 1 h at 37 °C and 5% CO_2_ with gentle agitation every 15 min. The virus inoculum was removed, and MEM 1% FBS was added to the CC and CV conditions. On the other hand, for the CW and CWV conditions, MEM supplemented with 1% FBS and the established diluted concentration of WDL (30 µM) was added. Cells were incubated for 24 h at 37 °C in a humidified atmosphere with 5% CO_2_. Culture media were removed, and cells were washed twice with cold phosphate-buffered saline (PBS) and harvested by trypsinisation, and a subsequent addition was made of cold MEM supplemented with 10% FBS for trypsin neutralisation and collection of cells. Centrifugation was performed, and the supernatant was removed and discarded. The cell pellets were then washed with cold PBS, followed by another centrifugation step and complete removal of the supernatant for subsequent HR-MAS ^1^H-NMR analysis.

### 4.8. HR-MAS ^1^H-NMR and 2D Analysis

All HR-MAS ^1^H NMR spectra were acquired on a Bruker Avance spectrometer (Bruker Biospin, Germany) operating at 400 MHz for ^1^H, equipped with a triple nuclei 4 mm g-HR-MAS 400 SB BL4 probe and performing at a magic-angle spinning frequency of 3.5 kHz and 298 K. Cell pellets were resuspended in 12 µL D_2_O (deuterated water) for the experiments. One-dimensional water-suppressed ^1^H-NMR spectra were acquired with the noesygppr1d (NOESY 1D) pulse sequence. T_2_-edited spectra were obtained using the cpmgpr1d (Carr–Purcell–Melboom–Gill, CPMG) pulse sequence, both with 256 transients and a recycling delay of 4 s. Two-dimensional ^1^H-^1^H total correlation spectroscopy (TOCSY) experiments were recorded using the mlevgpphw5 pulse sequence, ns = 110, mixing time = 60 ms, and the ^1^H-^13^C heteronuclear single quantum coherence (HSQC, hsqcetgpprsisp2.2 pulse sequence, ns = 384) experiments were also performed on randomly selected samples.

### 4.9. Data Preprocessing, Statistical Analysis and Metabolite Identification

The HR-MAS ^1^H-NMR cell spectra obtained were manually processed for phase and baseline corrections using MestReNova 12.0 software. Lactate (3H, δ 1.33, J = 7.0 Hz, doublet) was used as a chemical shift reference. The spectra were binned (0.005 ppm) and exported as csv matrices. Spectra regions below δ 0.5 and water region (δ 4.55–5.10) and above δ 8.7 were excluded from the analysis. Normalisation to a constant sum (100) of the spectra’s intensities was performed to reduce concentration differences. Assignment of the metabolites was based on chemical shifts, correlation assignments (TOCSY and HSQC) and databases, such as the Human Metabolome Database [80] and Biological Magnetic Resonance Data Bank [81].

All statistical analyses were performed using the MetaboAnalyst online platform [82]. Multivariate analyses were performed using the matrices with a total of 810 variables for the parts of the aliphatic regions of water suppression (noesy1d) and T_2_-edited (CPMG) spectra and constructed for all conditions (CC × CW × CWV × CV, with 24 spectra in total), as well as matrices containing each condition compared to the healthy control cells (CC), namely, CC × CW, CC × CWV, CC × CV, and also compared to the viral control (CV × CWV). Data were normalised by median and autoscaling.

Principal component analyses (PCA) were performed for all spectra (all conditions, separately for data obtained by the noesygppr1d and cpmgpr1d pulse sequences) to explore intrinsic groupings within samples, reduce data dimensionality and identify outliers [83]. Partial least squares discriminant analysis (PLS-DA) models were generated to classify samples from the four conditions evaluated together and compared to the healthy control (CC) and identify the essential variables [83,84]. Model validation was assessed by leave-one-out cross validation (LOOCV) to evaluate its accuracy and performance, and variable of importance in projection (VIP) scores were analysed to identify the most different chemical shifts among samples. The validated models’ accuracy, specificity and sensitivity values were computed [85]. Univariate analysis was performed to confirm the discriminant metabolites identified by multivariate analysis (PLS-DA) between two conditions (CC × CV; CC × CW; CC × CWV; CV × CWV) and for visualisation of the relative metabolite levels between the groups, measured as peak intensities, using a *t*-test with *p* < 0.05 as the threshold for statistically significant difference.

## 5. Conclusions

With recurrent epidemics worldwide and the risk of developing long-term complications in human hosts, the effects of arboviral infections by Chikungunya and other viruses are concerning. To date, no licensed antiviral drugs or vaccines are available. Although studies have provided numerous promising results in drug and vaccine development, research contributing to understanding viral mechanisms is crucial.

In this study, we aimed to explore the potential of HR-MAS ^1^H NMR to understand the metabolic effects of viral infection and viral inhibition by WDL with in vitro and metabolomic assays. Our results demonstrate that WDL inhibits viral nsP2 protease with considerably low IC_50_ values. At the same time, in cell culture, a much more complex environment, WDL slightly impairs the post-viral-entry steps in Vero E6 cells in a dose-dependent manner, where, perhaps, higher dosages of the molecule might have been more efficient in inhibiting viral replication in vitro, which were limitations in our study.

To date and to our knowledge, there are no HR-MAS NMR metabolomic studies regarding arboviruses of the *Togaviridae* family. HR-MAS NMR and chemometric analysis tools allowed us to trace a metabolic profile of Vero cells, along with metabolite disturbances triggered by Chikungunya virus infection, which mainly affected central energy, amino acid and phospholipid metabolism. We could also observe that WDL contributes to glucose synthesis in Vero cells. During infection, it might support anabolic pathways, lowering oxidative stress and cell damage, although it is speculative and provides further insights into future studies regarding these allegations.

## Figures and Tables

**Figure 1 ijms-25-01414-f001:**
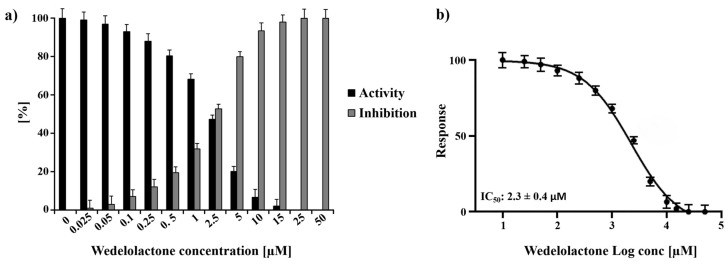
Normalised activity and inhibitory effects of WDL on nsP2 protease. (**a**) Inhibition of nsP2 by WDL with the concentrations ranging from 0 to 50 µM. (**b**) Dose response curve of WDL and nsP2 depicting the the half-maximal inhibitory concentration (IC_50_) of 2.3 ± 0.4 µM. The experiments were performed as triplicates and the results are shown as mean ± standard deviation (SD).

**Figure 2 ijms-25-01414-f002:**
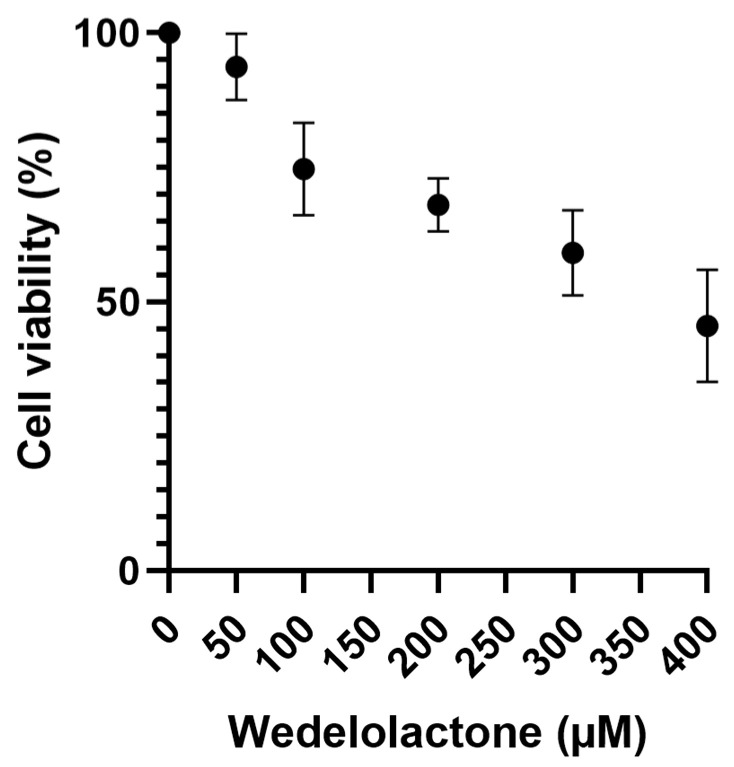
Vero E6 cell viability MTT assay using a range of WDL concentrations from 50 to 400 µM where live cells were quantified. Cell viability percentages are shown on the *y* axis, while WDL concentrations (µM) are on the *x* axis. Error bars represent standard deviations. Values are the mean ± standard error obtained from three independent experiments.

**Figure 3 ijms-25-01414-f003:**
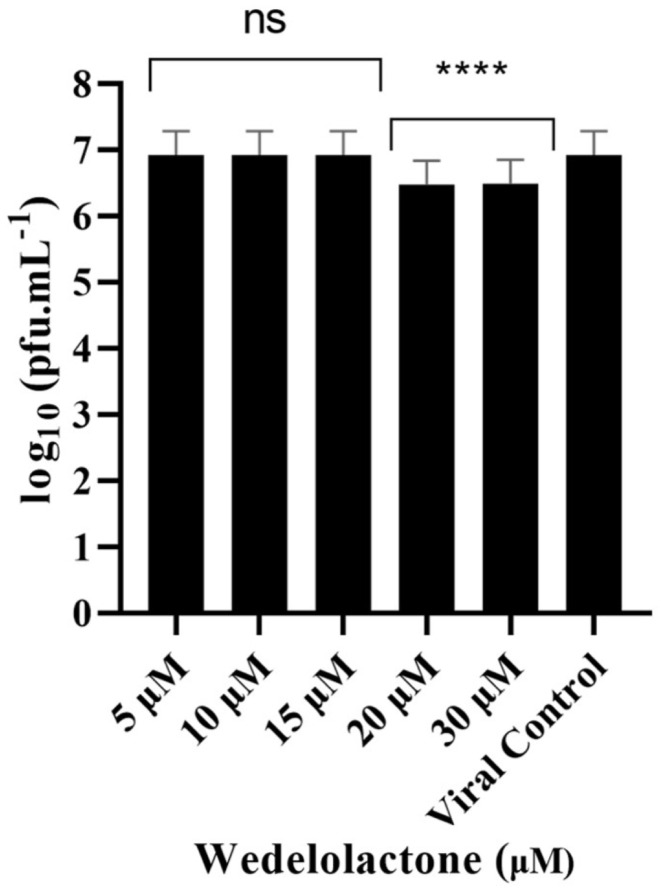
Dose-dependent inhibitory effects of WDL on CHIKV titre yields. Cells were infected with CHIKV at MOI of 0.1 and treated with different concentrations of WDL ranging from 5 µM to 30 µM. The reduction in the viral titre was determined by plaque-forming assay after collection and titration of the supernatants 24 h postinfection and treatment. PFU infectivity titration of CHIKV is shown on the left vertical axis. Error bars represent standard deviations. Values are the mean ± standard error obtained from three independent experiments. Asterisks indicate statistical significance between the control and each group as determined by two-way ANOVA and subsequent Dunnett’s test (ns, not significant; ****, *p* < 0.0001).

**Figure 4 ijms-25-01414-f004:**
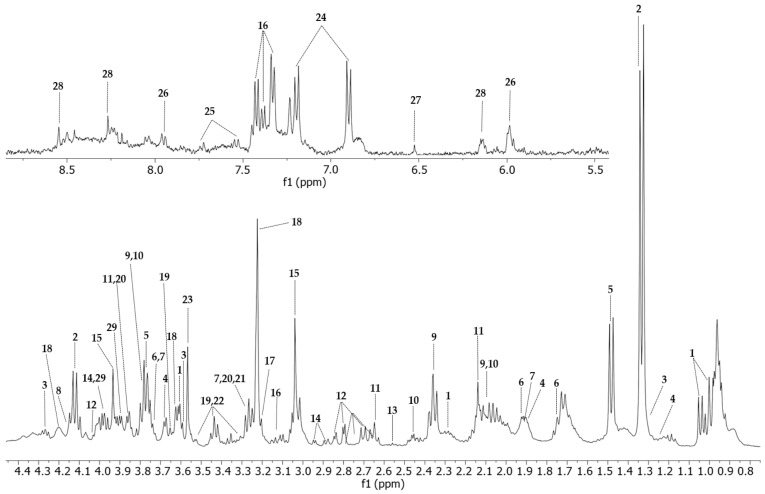
HR-MAS ^1^H-NMR T_2_-edited (CPMG) spectra (400 MHz) of all four conditions. Vero E6 control sample is represented in green (CC), cells treated with WDL (CW) are in blue, cells infected and treated with the molecule (CWV) are represented in purple and CHIKV-infected cells are represented in red (CV). Identified metabolites are Val (valine); Lac (lactate); Iso (isoleucine); Pro (proline); Met (methionine); Asp (aspartate); Cre (creatine); Phe (phenylalanine); Pho (phosphocholine); Glu (glucose); Bet (betaine); Myo (myo-inositol). HR-MAS ^1^H-NMR with T2 filter spectrum of a Vero E6 cell sample acquired with a Bruker Avance III 400 MHz using a CPMG (*cpmgpr1d*) pulse sequence. The upper image of the spectrum comprises the region between 5.0 and 9.00 ppm, amplified 20 times, and the lower image shows the 0.80 to 4.55 ppm region. All peaks assigned to a particular metabolite are numbered. 1: valine; 2: lactate; 3: threonine; 4: isoleucine; 5: alanine; 6: lysine; 7: arginine; 8: proline; 9: glutamate; 10: glutamine; 11: methionine; 12: aspartate; 13: hypotaurine; 14: asparagine; 15: creatine; 16: phenylalanine; 17: choline; 18: phosphocholine; 19: glucose; 20: betaine; 21: trimethylamine-N-oxide; 22: myo-inositol; 23: glycine; 24: tyrosine; 25: tryptophan; 26: UDP-N-acetylglucosamine, UDP-glucose, UDP-glucuronate; 27: fumarate; 28: ATP; 29: serine.

**Figure 5 ijms-25-01414-f005:**
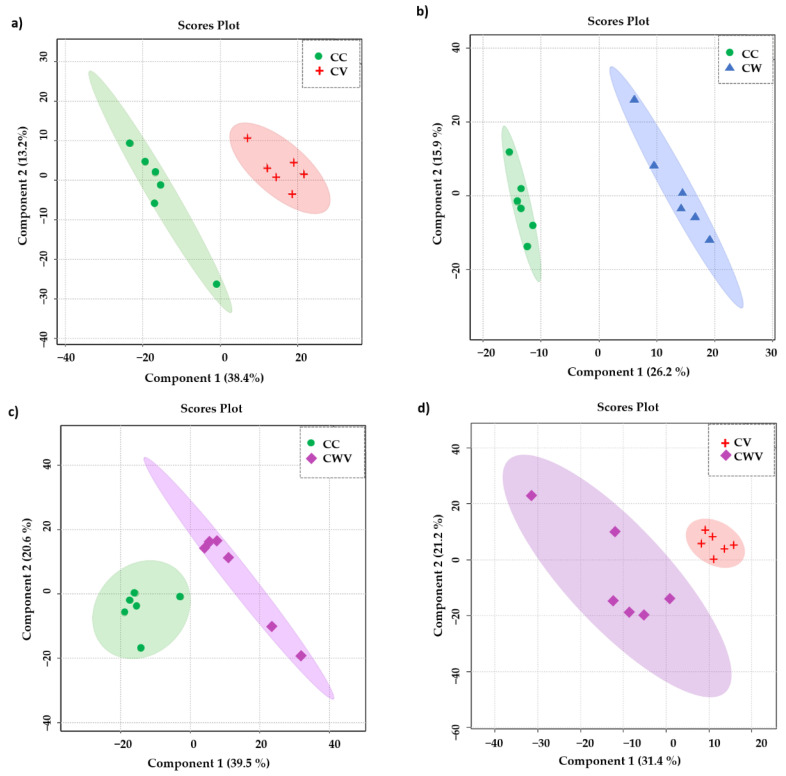
PLS-DA scores plot in 2D of the HR-MAS ^1^H-NMR data showing the discrimination between the healthy cell control (CC, green circles) and the other conditions: (**a**) CC × CV (red crosses); (**b**) CC × CW (blue triangles); (**c**) CC × CWV (purple diamonds); (**d**) CV × CWV. The explained variance of the models with the two first latent variables is represented on the *x* and *y* axes.

**Figure 6 ijms-25-01414-f006:**
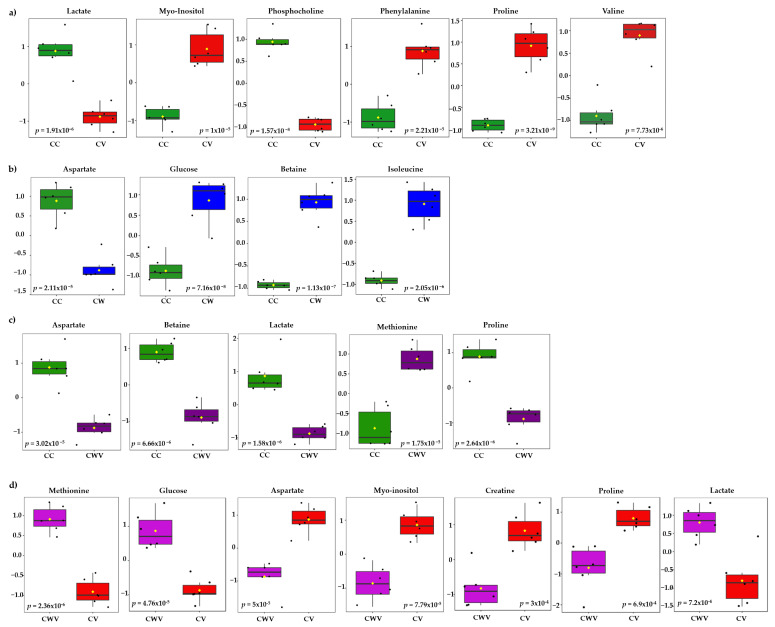
Box plots of discriminant metabolites identified in the univariate analysis (*t*-test, *p* < 0.05) of the (**a**) CC × CV, (**b**) CC × CW, (**c**) CC × CWV, (**d**) CV × CWV groups. Relative metabolite levels between the groups are shown (measured as peak intensities). The black dots represent metabolite levels in the samples, and the yellow diamond represents the average value for the group, as indicated on the y axis. Groups are CC, green; CV, red; CW, blue; CWV, purple.

**Figure 7 ijms-25-01414-f007:**
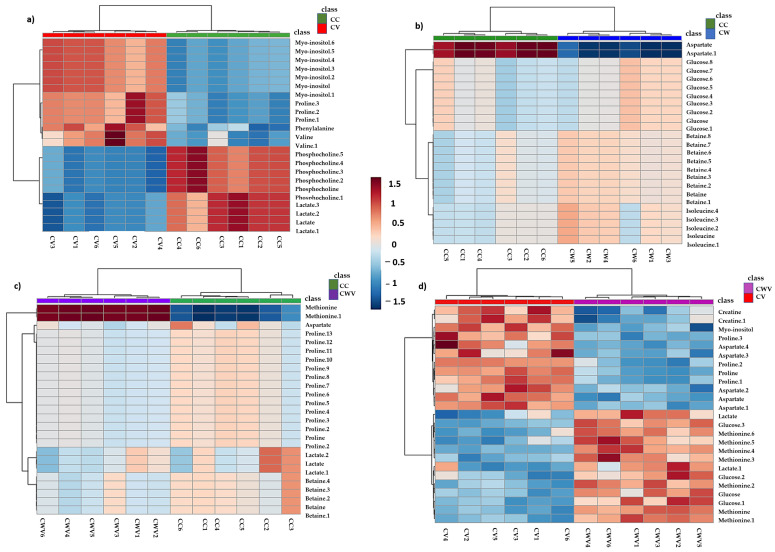
Hierarchical clustering through heat maps analysis using Euclidean distance. The lines represent the discriminant variables from the PLS-DA analysis for (**a**) CC × CV, (**b**) CC × CW, (**c**) CC × CWV and (**d**) CV × CWV. The columns represent the samples, and the colour bars on the right represent the relative concentrations of the discriminant metabolites (red: higher relative concentration; blue: lower relative concentration).

**Figure 8 ijms-25-01414-f008:**
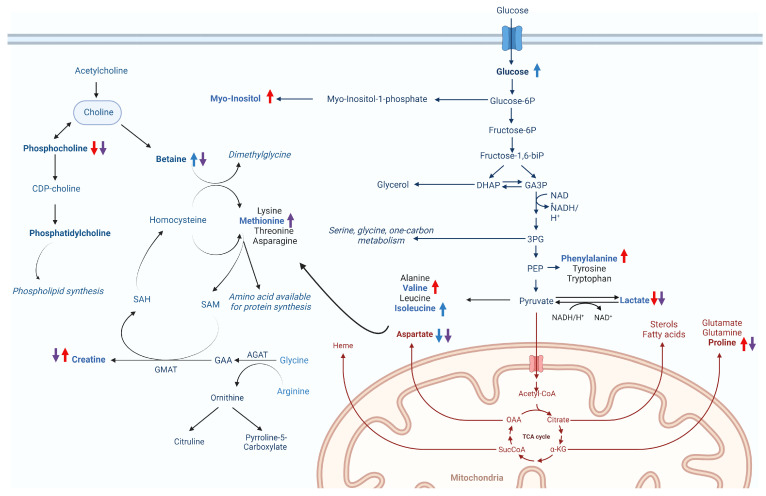
Hypothesised changes to metabolic pathways in Vero E6 cells caused by CHIKV infection and/or WDL treatment. The coloured arrows indicate increased (upwards) or decreased (downwards) metabolite levels for the different conditions analysed: CV (red), CW (blue) and CWV (purple).

**Table 1 ijms-25-01414-t001:** Accuracy, R^2^ and Q^2^ values of the best performance PLS-DA models calculated by LOOCV.

Classes	Accuracy	R^2^	Q^2^
CC × CV	0.91	0.99	0.78
CC × CW	1.0	0.99	0.90
CC × CWV	1.0	0.99	0.85
CV × CWV	0.91	0.99	0.72
CC × CV × CW × CWV	0.87	0.98	0.85

## Data Availability

The data presented in this study are available in the article and its supplementary material.

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
