# Peer review of "HR-MAS NMR Metabolomics Profile of Vero Cells under the Influence of Virus Infection and nsP2 Inhibitor: A Chikungunya Case Study"

_ijms, 2024, doi:10.3390/ijms25031414_

Round 1

Reviewer 1 Report

Comments and Suggestions for Authors

Chikungunya virus (CHIKV) is a mosquito-borne arbovirus that causes fever and joint pain in humans. While Chikungunya is rarely fatal, its impact lies in the devastating nature of the joint pain, which can persist for weeks or months and significantly affect the quality of life for those infected. Currently, there are no approved antiviral drugs available that specifically target the Chikungunya virus and the treatment focuses on managing symptoms, mostly with pain relievers. Hence, more research is needed to develop effective antiviral drugs against CHIKV. The present study explored the potential of Wedelolactone (WDL), a compound from Eclipta prostate, in inhibiting nsP2 of CHIKV and the subsequent reduction of the virus, as well as the effects of CHIKV infection and WDL on Vero cell metabolomics using NMR-based metabolomics. Although the study is very important considering the need for potent antiviral drugs in treating CHIKV, several points need to be addressed prior to its publication.

1.   While proven from the current study itself, WDL showed a slight antiviral effect in CHIKV infected Vero cells, authors should justify how it became relevant to check the NMR Metabolomics profile of a rather not-so-potent antiviral candidate

2.   Adding to point 1, the study revealed no difference in metabolic alterations, except for creatine, between CWV and CV. The candidate molecule could only affect the creatine level? Probably is this due to the feeble inhibition of CHIKV? The authors should clarify and discuss whether this limited impact on metabolites, aside from creatine, is indicative of the candidate's weak inhibition against CHIKV.

3.   The authors describe the in vitro inhibition of CHIKV nsP2 as "moderately strong." It is crucial for the authors to provide a comparative standard to justify this assessment (see line 99).

4.   Quantity of inhibition in molar terms needs to be stated as 100% inhibition corresponds to 10µM CHIKV nsP2pro inhibition.

5.   In Figure 1 (a), as inhibition is relatively calculated, activity will be obviously reciprocal to the inhibition. Depicting both activity and inhibition in the graphs appears redundant. The quantity of inhibition may be declared in the graph. 

6.   Figure 1 lacks a statement regarding the error in the graph and how values are normalized.

7.   The cytotoxicity assessment of WDL reveals a CC50 value of 373.5µM with an error of 154.2µM. The error appears to be relatively large, stretching the range of CC50 value. Hence, the experiment needs to be repeated with caution to reduce the error to obtain a more precise CC50 value.

8.   In Figure 2, the error bar corresponding to the 200µM WDL appears to be relatively small and seems incongruent with the error value stated previously in the result. 

9.   In the anti-viral activity assessment of WDL, please state the logical rationale behind selecting a maximal concentration of 30µM for assessment while the CC50 value is more than 10times higher. 

10.Figure 3, on its own, does not effectively convey the actual inhibition and half maximal inhibitory concentration. Including a percentage of the inhibition graph would improve visualization.

11.The contradiction between the concentration of IC50 of nsP2 inhibition and IC50 of anti-viral activity of WDL necessitates the bioavailability assays to be performed to get better clarity.

12.In lines 336-338 “We speculate that the decreased levels observed compared to the CV samples might be due to the reduced energetic demands with the lower viral progeny yields “, authors should justify how only creatine level were affected but not other metabolites level due to the lower viral progeny yields.

Comments on the Quality of English Language

The manuscript requires thorough proofreading to correct grammatical errors. Abbreviations should be written in their expansion form at their first instance of usage in the manuscript.

Author Response

Comment 1

While proven from the current study itself, WDL showed a slight antiviral effect in CHIKV infected Vero cells, authors should justify how it became relevant to check the NMR Metabolomics profile of a rather not-so-potent antiviral candidate

Response 1: Thank you for your constructive feedback. Although Wedelolactone (WDL) exhibited moderate antiviral effects in CHIKV-infected Vero cells, its prominence lies in the significant inhibition of the NS2 protease. This notable inhibitory activity warrants the NMR metabolomics analysis, as substantial metabolic alterations were anticipated to accompany such inhibition. Importantly, WDL demonstrated no cytotoxic effects, underscoring its potential as an antiviral agent. Thus, metabolomics analysis is essential to comprehend the biological implications of WDL and its interaction with CHIKV, despite its seemingly low antiviral potency.

Comment 2   Adding to point 1, the study revealed no difference in metabolic alterations, except for creatine, between CWV and CV. The candidate molecule could only affect the creatine level? Probably is this due to the feeble inhibition of CHIKV? The authors should clarify and discuss whether this limited impact on metabolites, aside from creatine, is indicative of the candidate's weak inhibition against CHIKV.

Response 2: Within the CVxCWV analyses, notable variations in various metabolites emerged, including decreased levels of myo-inositol, proline, creatine, and aspartate, alongside elevated concentrations of glucose, lactate, and methionine in the CWV group when compared to infected cells (CV). Although we centered our discussion around creatine’s decrease, as it was unique to this condition and was not significant in the other analysed scenarios, we did observe differences that may support our hypothesis of reduced viral titer and energy demand, such as decreased myo-inositol (known to be increased in the CV samples possibly related to production of phosphatidylinositol that supports VRCs and replication) as well as proline (used for protein synthesis and energy production in cells). Increased levels of glucose, lactate and methionine and decreased aspartate is similar to the effects of WDL in cells (CW) possibly aiding energy production even during viral infection. This discussion has been added to the manuscript (lines 365-370).

Comment 3

The authors describe the in vitro inhibition of CHIKV nsP2 as "moderately strong." It is crucial for the authors to provide a comparative standard to justify this assessment (see line 99).

Response 3

We added IC50 values of small molecules and flavonoids with a reported inhibition effect against CHIKV nsP2 and compared these values with our results.

Hesperetin (2.5 µM) and Hesperidin (7.1 µM) [Eberle, R.J.; Olivier, D.S.; Pacca, C.C.; Avilla, C.M.; Nogueira, M.L.; Amaral, M.S.; Willbold, D.; Arni, R.K.; Coronado, M.A. In vitro study of Hesperetin and Hesperidin as inhibitors of zika and chikungunya virus proteases. PLoS ONE 2021, 16, e0246319.],

1,3-Thiazolbenzamide derivatives (Between 8.3 and 13.1 µM) [Ivanova, L.; Rausalu, K.; Žusinaite, E.; Tammiku-Taul, J.; Merits, A.; Karelson, M. 1,3-Thiazolbenzamide Derivatives as Chikungunya Virus nsP2 Protease Inhibitors. ACS Omega 2021, 6, 5786–5794.]

We removed moderately strong and added the following sentence, "This value is in the same range as previously reported flavonoids and small molecules targeting CHIKV nsP2; Hesperetin (2.5 µM) and Hesperidin (7.1 µM) [38], 1,3-Thiazolbenzamide derivatives (between 8.3 and 13.1 µM) [39].".

Comment 4

Quantity of inhibition in molar terms needs to be stated as 100% inhibition corresponds to 10µM CHIKV nsP2pro inhibition.

Response 4

We wrote the corresponding molar concentrations to each inhibition effect in figure 1A and were wondering because 100% inhibition corresponds to a WDL concentration of 25 µM.

Comment 5

In Figure 1 (a), as inhibition is relatively calculated, activity will be obviously reciprocal to the inhibition. Depicting both activity and inhibition in the graphs appears redundant. The quantity of inhibition may be declared in the graph.

Response 5

We changed the figure accordingly.

Comment 6  

Figure 1 lacks a statement regarding the error in the graph and how values are normalized.

Response 6

We added the missing information in the figure caption and in the material and method section.

Comment 7

The cytotoxicity assessment of WDL reveals a CC50 value of 373.5µM with an error of 154.2µM. The error appears to be relatively large, stretching the range of CC50 value. Hence, the experiment needs to be repeated with caution to reduce the error to obtain a more precise CC50 value.

Response 7: We appreciate your observation and understand the concern regarding the accuracy of the CC50 value, which was calculated as 373.5µM with an error of 154.2µM. However, we want to emphasize that we conducted all experiments with scientific rigor and appropriate design, ensuring the reliability of the obtained data. To facilitate understanding and accessibility of the results, we have included in the supplementary materials section Table S1 which presents the mean percentage of cell viability for all tested concentrations, along with standard deviations (ranging from 400 to 50 µM). Furthermore, we highlight that even considering the lowest possible value for CC50, according to our calculation (373.5 µM - 154.2 µM = 219.3 µM), this value is at least seven times higher than that used in our antiviral tests (30 µM). The viability assay for the 50 µM concentration showed a percentage of 93.6±10, with a 95% confidence interval, demonstrating an absence or low cytotoxicity at this concentration, which is substantially higher than that used in subsequent tests (30 µM). We publish cytotoxicity data in table (published in Viruses-MDPI and IJMS-MDPI) in the same manner as Table S1 that we have now added: https://www.mdpi.com/1999-4915/15/2/494 and https://www.mdpi.com/1422-0067/24/4/3333. We assert that the values presented in this manuscript are reliable and supported by robust data. We encourage consulting Table S1 for more details on the results. Thank you again for the critical review, and we are available for any further clarifications.

Comment 8

In Figure 2, the error bar corresponding to the 200µM WDL appears to be relatively small and seems incongruent with the error value stated previously in the result.

Response 8: Thank you for bringing attention to Figure 2. The error bar associated with the 200 µM WDL is indeed smaller due to the close values of replicates in the experiments. This consistency generally indicates robust results and does not raise substantial concerns. We chose not to modify this representation in the figure as it accurately reflects the real variability of the obtained data. Additionally, for enhanced transparency, we have included Table S1 in the supplementary materials. This table provides a more comprehensive view and facilitates access to the data, contributing to a clearer understanding of the results. We appreciate your scrutiny and remain available for any further clarifications.

Comment 9

In the anti-viral activity assessment of WDL, please state the logical rationale behind selecting a maximal concentration of 30µM for assessment while the CC50 value is more than 10times higher.

In response to your query regarding the selection of a maximum concentration of 30µM for the antiviral activity assessment of Wedelolactone, we would like to clarify that this decision was based on its significant efficacy observed in inhibiting the ns2 (nsP2) protease of CHIKV. This concentration was chosen to explore the antiviral potential of Wedelolactone, considering its safety profile observed at tested concentrations. We have included this information in the manuscript to elucidate the rationale behind our choice .

Comment 10

Figure 3, on its own, does not effectively convey the actual inhibition and half maximal inhibitory concentration. Including a percentage of the inhibition graph would improve visualization.

To enhance reader comprehension and visualization, we refrained from including the percentage of inhibition and opted to present inhibition results using logarithmic reduction, a commonly employed approach in literature. This choice was made considering that the inhibition results signify a decrease of nearly half a logarithmic unit in viral control titers, originally equating to approximately 7 logarithmic units.

Comment 11

The contradiction between the concentration of IC50 of nsP2 inhibition and IC50 of anti-viral activity of WDL necessitates the bioavailability assays to be performed to get better clarity.

The observed discrepancy between the IC50 values for nsP2 inhibition and the antiviral activity of WDL can be attributed to the complexity of in vitro assays in cell cultures, which are significantly more complex than protein-inhibitor interaction analyses. This added complexity might influence the observed efficacy of WDL, leading to the differences in IC50 values. While bioavailability assays could enhance our comprehension of WDL's antiviral activity, executing such experiments within the current timeframe proves unfeasible. Nevertheless, despite utilizing lower concentrations in this study, we successfully observed discernible variances in cells' metabolic responses during treatment, both with and without prior infection. Our primary objective was to track metabolic alterations induced by the virus when coupled with the inhibitor in the culture medium. Despite the modest inhibition results, metabolomics analysis still revealed changes that substantiate WDL uptake and metabolization by cells.

Comment 12

In lines 336-338 "We speculate that the decreased levels observed compared to the CV samples might be due to the reduced energetic demands with the lower viral progeny yields ", authors should justify how only creatine level were affected but not other metabolites level due to the lower viral progeny yields.

Explanation provided in Comment 2.

Comment 13

Comments on the Quality of English Language

The manuscript requires thorough proofreading to correct grammatical errors. Abbreviations should be written in their expansion form at their first instance of usage in the manuscript.

Response: Proofreading and correction of grammatical errors along with improvement of phrases in the discussion for clearer comprehension was performed accordingly.

Reviewer 2 Report

Comments and Suggestions for Authors

Present work highlights the nsP2 inhibitor is potential therapeutic agent against viral infection (CHIKV) by modulating the metabolic pathway. Here authors showed  how the metabolic signature is changed upon nsP2 protease inhibitor, Wedelolactone (CWV) which inhibits replication of virus. The positive aspects of this study is most of the results are convincing. However, there is lacking in the conceptual representation as inhibition of viral replication is due to interferon signature which is the part of innate immune pathway. Is there any the interferon response upon the nsP2 protease inhibitor, Wedelolactone (CWV) treatment and the metabolic signature is due to interferon response. This little bit confusing and more clarification needed for readers. Rest of the things are convincing and acceptable.  Below are some comments to clarify and make this work in better presentable format. 

Minor points

1.     Figure-2: Mention 50um in the X-axis.

2.     Figure-4: Mention the colour coding symbols in the figure itself what is what apart from the legends for better understanding. 

3.     Figure-5 and 7: Replace the figures with high resolution picture. 

Comments on the Quality of English Language

Minor editing of English is required. 

Author Response

Present work highlights the nsP2 inhibitor is potential therapeutic agent against viral infection (CHIKV) by modulating the metabolic pathway. Here authors showed  how the metabolic signature is changed upon nsP2 protease inhibitor, Wedelolactone (CWV) which inhibits replication of virus. The positive aspects of this study is most of the results are convincing. However, there is lacking in the conceptual representation as inhibition of viral replication is due to interferon signature which is the part of innate immune pathway. Is there any the interferon response upon the nsP2 protease inhibitor, Wedelolactone (CWV) treatment and the metabolic signature is due to interferon response. This little bit confusing and more clarification needed for readers. Rest of the things are convincing and acceptable.  Below are some comments to clarify and make this work in better presentable format. 

We appreciate your insightful observation. Despite the absence of interferon genes in Vero E6 cells and, consequently, their inability to mount an interferon response to infection, our aim was to examine how these cells respond to infection within these constraints. Additionally, we sought to unveil their metabolic profile for the first time, as no such information is currently available in the literature to our knowledge.

While we did not specifically investigate whether WDL stimulates an interferon response, we postulate that it is unlikely given the absence of interferon genes in these cells. In addition to WDL's recognized inhibitory potential against nsP2, we propose that its inhibitory activity in the cellular environment stems from its potential antioxidant capacity and its capacity to supply energy to the cells.

Minor points

  1. Figure-2: Mention 50um in the X-axis.

The 50uM point was added to the figure.

  1. Figure-4: Mention the colour coding symbols in the figure itself what is what apart from the legends for better understanding. 

Figure 4 was changed to Figure S1 in the supplementary material and colors were explained in the legend.

  1. Figure-5 and 7: Replace the figures with high resolution picture. 

Figures resolution was improved and replaced.

Reviewer 3 Report

Comments and Suggestions for Authors

The paper   reports results of a metabolomic  study profiling  by  HR-MAS NMR the mammalian cells upon  Chikunguya virus infection and the effest of an inhibitor.

Tom y esperienze it is the first example of the use of this approach to the biochemical effects of the viral  infection and about thev ecffect of an inhibitor.

The spectra are really beatiful and the resultss discussed are interesting. Thus I recomend the paper for the publication.

Only few observations for a less familial reader interiste to expand this approach to viral inhibition.

I suggest that  the spectrum shown in S1 shift back in the main text can help to introduce better to the protocols applied  introducing simply  the Fig 1 where the filtering by  T2 simplifies the spectrum ( but decreasing intensities).

Secondly a short general introduction to the inhibitor  Wedelolactone may help to know better this molecule.

Thirdly the statistical analysis give important information of the effect of the infection and the inhibition. A very short  meaning of the differe3nt approach may help the less familial reader to this important procedure and abbout the importance of the results obtained.  This would reinfeorce the sentence :” We  suggest that metabolomics studies of viruses and inhibitors may represent a rapid and

efficient approach for unveiling viral and inhibitor mechanisms that serve as the basis for  further investigation of metabolic pathways and drug development through omics

integration.”

Particular interesting the biochemical  evidences obtained  about  “changes in the levels of lactate, Myo-inositol, phosphocholine, glucose, betaine, and a few specific

amino acids.

Effectively as the AAs say “HR-MAS NMR and chemometric analysis  tools allowed to trace a metabolic profile of Vero cells, along with metabolite  15 of 20

disturbances triggered by Chikungunya virus infection, which mainly affected central energy, amino acid and phospholipid metabolism.”

Author Response

I suggest that the spectrum shown in S1 shift back in the main text can help to introduce better to the protocols applied  introducing simply  the Fig 1 where the filtering by  T2 simplifies the spectrum ( but decreasing intensities).

Response: Thank you for your suggestion. Figure S1 was shifted back into the main manuscript and the CPMG spectra with the most relevant metabolites in the statistical analyses was put into the supplementary material.

Comment 2

Secondly a short general introduction to the inhibitor Wedelolactone may help to know better this molecule.

Response: Lines 66-71 in the introduction contain a brief description of WDL, as well as its identified functions in several scientific papers.

Comment 3

Thirdly the statistical analysis give important information of the effect of the infection and the inhibition. A very short meaning of the different approach may help the less familial reader to this important procedure and about the importance of the results obtained. This would reinfeorce the sentence :" We  suggest that metabolomics studies of viruses and inhibitors may represent a rapid and efficient approach for unveiling viral and inhibitor mechanisms that serve as the basis for  further investigation of metabolic pathways and drug development through omics integration."

Response: Thank you for pointing these observations. Short descriptions of the statistical methods used are introduced in the text to aid readers comprehension which can be read at lines 463-476.

Comment 4

Particular interesting the biochemical  evidences obtained  about "changes in the levels of lactate, Myo-inositol, phosphocholine, glucose, betaine, and a few specific amino acids. Effectively as the AAs say "HR-MAS NMR and chemometric analysis  tools allowed to trace a metabolic profile of Vero cells, along with metabolite  15 of 20 disturbances triggered by Chikungunya virus infection, which mainly affected central energy, amino acid and phospholipid metabolism."

Response: We very much appreciate your interest in our results.

Reviewer 4 Report

Comments and Suggestions for Authors

The work “HR-MAS NMR Metabolomics profile of Vero cells under the influence of virus infection and nsP2 inhibitor: A Chikungunya Case Study by Rafaela dos S. Peinado and cols is a good work of NMR based metabolomic analysis. It is well written and clear however, at this stage there are some concerns that should be addressed.

1.     The most interesting fitting is Infected cells vs Infected and treated cells (Figure 5 d of the manuscript) and to know if infected cells once treated return to “control” state. For this, the model shown in Fig5d could be used to classify control samples.

2.     There are only 6 samples per condition, while this number is usual in literature reports it may be a bit low to achieve statistical significance thus should provide statistical relevance of their PLS-DA models, ideally CV-ANOVA or p values provided by Metaboanalyst.

3.     Data pre processing. Line 440 states that normalization was done to “constant sum (100)” while in line 451 “Data were normalised by median”. Please clarify. More important in line 451 it is also stated that data was “auto-scaled”. Usually paretto scaling is used; authors should use paretto scaling.

4.     Figure 6, Box plots. What do the box plots correspond to? Is it the area under the curve (integral or deconvolution) for that metabolite or the intensity of the bin that appears close to the peak? Does it come from bin intensity (metabolic fingerprint) or identified metabolite area (metabolic profiling)?

Author Response

The most interesting fitting is Infected cells vs Infected and treated cells (Figure 5 d of the manuscript) and to know if infected cells once treated return to "control" state. For this, the model shown in Fig5d could be used to classify control samples.

Response: We appreciate this comment, however, we did not understand this suggestion. We performed all different conditions compared to the healthy control cells (CC) and to the viral infected cells (CV). The model shown in Figure 5d represents the infected group (CV) compared to the treated-infected group (CWV). We also generated the model of CC compared to CV, as well as to CWV.

Comment 2

There are only 6 samples per condition, while this number is usual in literature reports it may be a bit low to achieve statistical significance thus should provide statistical relevance of their PLS-DA models, ideally CV-ANOVA or p values provided by Metaboanalyst.

Response: The PLSDA models were validated by LOOCV and results from all comparisons are presented in Table 1. The univariate analyses with a t-test or ANOVA presented p values which are depicted in all boxplots for each metabolite identified as significant (p < 0.05) in Figure 6.

Comment 3

Data pre processing. Line 440 states that normalization was done to "constant sum (100)" while in line 451 "Data were normalised by median". Please clarify. More important in line 451 it is also stated that data was "auto-scaled". Usually paretto scaling is used; authors should use paretto scaling.

Response: Thank you for you substantial feedback. As we understand, data normalization can be performed at several stages of analysis. We first performed normalization on spectra in the pre-processing stages (where spectra were normalized to a constant sum) to reduce the effects from higher peak intensities/concentrations. Next, normalization by median and autoscaling were performed only when data matrices were uploaded into MetaboAnalyst for statistical and chemometric analysis. We tested several normalization conditions, and auto-scaling was chosen as the best approach for our data because it provided results closer to a normal distribuition of data, while pareto scaling did not. These processing stages can vary depending on data distribution and results obtained, it is not a rule to perform pareto scaling in all anayses.

Comment 4

Figure 6, Box plots. What do the box plots correspond to? Is it the area under the curve (integral or deconvolution) for that metabolite or the intensity of the bin that appears close to the peak? Does it come from bin intensity (metabolic fingerprint) or identified metabolite area (metabolic profiling)?

Response: The boxplots derived from univariate analyses correspond to bin intensity (ppm) which gives us a general idea of relative metabolite concentrations and whether it decreases or increases in the conditions analysed.

Round 2

Reviewer 1 Report

Comments and Suggestions for Authors

Thank you for addressing most of the raised concerns and providing explanations.